

# Two-headed butterfly vs. mantis: do false antennae matter?

Tania G. López-Palafox[1] and Carlos R. Cordero[2]

[1] Posgrado en Ciencias Biológicas, Universidad Nacional Autónoma de México, Ciudad de México, México
[2] Departamento de Ecología Evolutiva, Instituto de Ecología, Universidad Nacional Autónoma de México, Ciudad de México, México

## ABSTRACT

The colour patterns and morphological peculiarities of the hindwings of several butterfly species result in the appearance of a head at the rear end of the insect's body. Although some experimental evidence supports the hypothesis that the "false head" deflects predator attacks towards the rear end of the butterfly, more research is needed to determine the role of the different components of the "false head". We explored the role of hindwing tails (presumably mimicking antennae) in predator deception in the "false head" butterfly *Callophrys xami*. We exposed butterflies with intact wings and with hindwing tails experimentally ablated to female mantises (*Stagmomantis limbata*). We found no differences in the number of butterflies being attacked and the number of butterflies escaping predation between both groups. However, our behavioural observations indicate that other aspects of the "false head" help *C. xami* survive some mantis attacks, supporting the notion that they are adaptations against predators.

## INTRODUCTION

Butterfly wings are canvases on which evolution designs solutions to the problems posed by thermoregulation, sexual selection and predation (*Monteiro & Prudic, 2010*; *Kemp & Rutowski, 2011*). These adaptations frequently involve compromises between selective pressures when optimal trait values differ between functions (*Ellers & Boggs, 2003*), although sometimes they coincide (*Finkbeiner, Briscoe & Reed, 2014*). Several butterfly species exhibit colour patterns and morphological peculiarities in their hindwings that suggest, at least to the human eye, that a butterfly resting with its wings closed possess a second head at the rear end of its body (*Robbins, 1980*; *Cordero, 2001*). This appearance is enhanced by peculiar behaviours, such as the back and forth movements of the closed hindwings that presumably permit the "false antennae"—the "tails" frequently present in the border of the anal angle of the hindwings (Fig. 1A)—mimic the movements of real antennae (*Robbins, 1980*; *López-Palafox, Luis-Martínez & Cordero, 2015*). False head butterflies are especially common among the subfamily Theclinae (Lycaenidae). Several specific hypotheses on the function of the "false head" have been advanced; all of them consider visually oriented predators as the main selective pressure, and avoidance or deflection of attacks as the main advantage (*Robbins, 1980*; *Cordero, 2001*). Although false head butterflies are textbook

Corresponding author
Carlos R. Cordero,
crafaelcm@gmail.com,
cordero@ecologia.unam.mx

examples of anti-predator adaptations (e.g., *Wickler, 1968*; *Ruxton, Sherratt & Speed, 2004*), to the best of our knowledge, there are only two published experimental studies testing the effect of false heads on probability of predation in live butterflies.

*Sourakov (2013)* exposed two *Calycopis cecrops* (Lycaenidae) butterflies, a species with false head, and thirteen individuals from eleven species of butterflies and moths without false heads, to one individual predatory salticid spider (*Phidippus pulcherrimus*). The spider repeatedly failed to trap the lycaenid butterflies because it directed all its attacks towards the false head, but captured all individuals from the other species, mostly (11 out of 13 cases) in the first or second attack. *Wourms & Wasserman (1985)* added artificial "false heads" to *Pieris rapae* (Pieridae) butterflies by attaching tails ("false antennae") and painting spots ("false eyes") on the anal angle of the hindwings, as well as by painting lines converging on the anal angle, three of the main components of false heads identified by *Robbins (1980)*. *Wourms & Wasserman (1985)* compared predation rates by Blue Jays (*Cyanocitta cristata*) between intact butterflies and butterflies with false heads added. All control and experimental butterflies attacked were caught, but the percentage of butterflies escaping during handling was twice as large in the treatment with artificial false heads as in the control group (16 out of 60 *vs.* 10 out of 79, respectively). The authors mention that butterflies escaped due to "mishandlings" by the birds, i.e., due to errors resulting from misdirected strikes while handling captured prey (*Wourms & Wasserman, 1985*). Thus, the experimental research available supports the idea that false heads help butterflies to deflect attacks away from their less vulnerable end (*Wourms & Wasserman, 1985*; *Sourakov, 2013*).

However, these experimental studies have some limitations. *Sourakov*'s (*2013*) sample size was very small and the control group differed in a number of morphological and behavioural aspects besides the absence of a false head. *Wourms & Wasserman (1985)* recognized that the wing shape of *P. rapae* is different from that of "false-head" Lycaenidae and that some of the behaviours associated with the functioning of false heads are absent in this species. Furthermore, although these studies support the deflecting function of false heads, visually guided predators of butterflies exhibit a variety of sensory capabilities and employ different hunting strategies, and it is not clear if false heads are useful against all them.

Salticid spiders and birds are active hunters that are probably able to use fine details to identify and attack butterflies, while sit-and-wait predators, such as mantises, appear to recognize prey by assessing a number of general features in objects found in the environment (*Kral, 2012*; *Prete et al., 2013*). According to behavioural and electrophysiological studies (reviewed in *Prete et al., 2013*), the main features used by mantises include the size of the object, contrast with the background, leading edge length, speed and movement pattern. Thus, considering the last two features, we hypothesize that the movement of false antennae (i.e., the "tails" present in the border of the anal angle of the hindwings) deflects mantis attacks to a less vulnerable area and increases the probability of escape. We tested this idea by measuring the effects of experimentally ablating the hindwing tails of the false head butterfly *Callophrys xami* (Lycaenidae: Techlinae) (Fig. 1) on the probability of exhibiting hindwing back-and-forth movement, and on the probability of being attacked and captured by female mantises (*Stagmomantis limbata*).
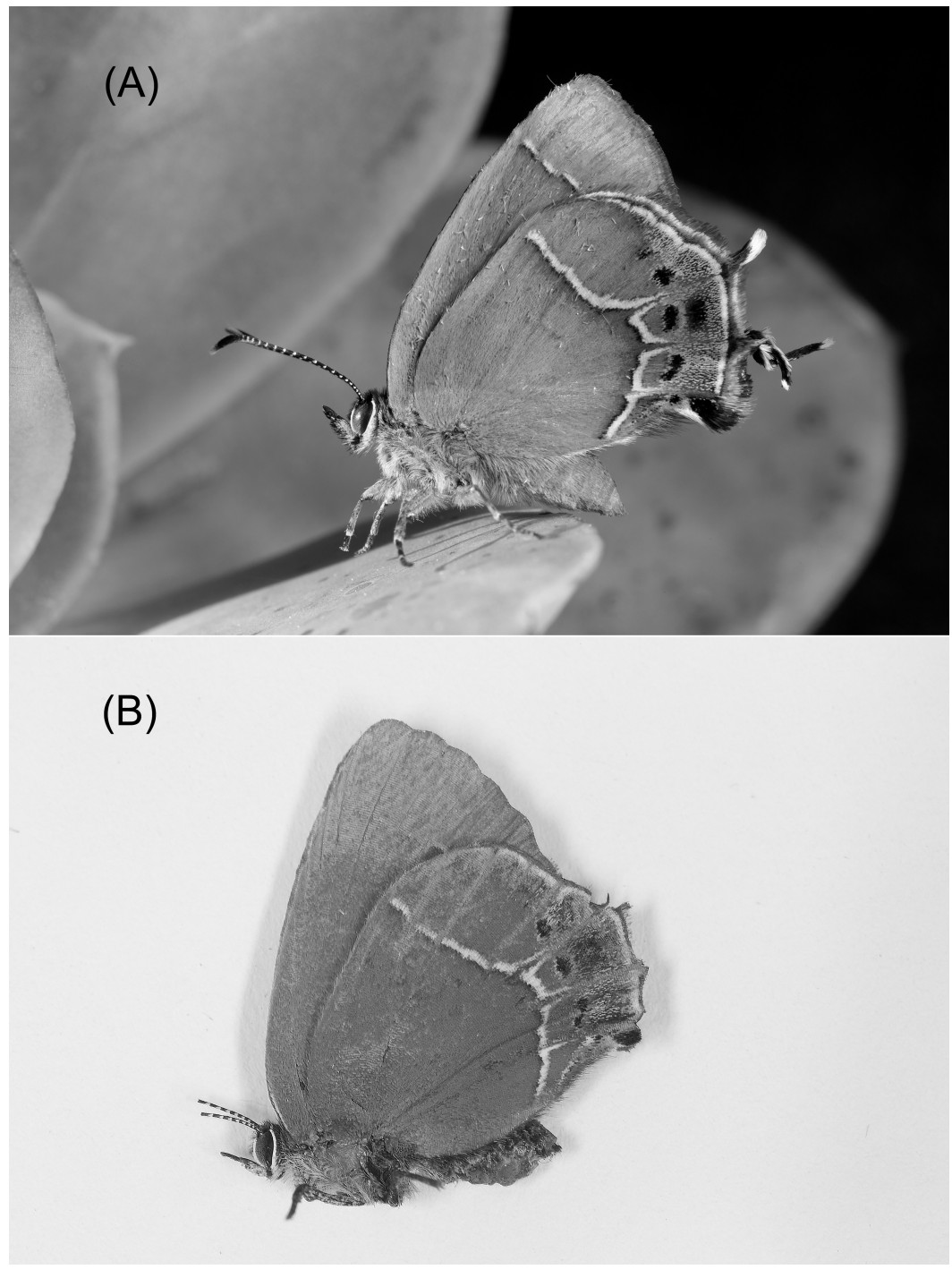

**Figure 1** *Callophrys xami* **(A) with hindwing tails intact (control) and (B) with hindwing tails experimentally ablated (dead experimental specimen with broken antennae).** Photographs by Raúl Iván Martínez.

## MATERIALS AND METHODS

Experimental butterflies were raised from eggs laid by three females collected in the Pedregal de San Ángel Ecological Reserve (PSAER) of the Universidad Nacional Autónoma de México (UNAM), located in the main campus of the UNAM in the South of Mexico City. *Callophrys xami* is a multivoltine "false head" butterfly whose main food plant in the collection site is *Echeveria gibbiflora* DC (Crassulaceae). Rearing methods followed *Jiménez & Soberón (1988–1989)*.

The predators used in the experiment were adult females (males did not attack butterflies in pilot tests) of the mantis *Stagmomantis limbata*, a species living in the PSAER and, therefore, a potential natural predator of *C. xami*. Some of the females were the offspring of a female collected in the PSAER, whose nymphs were maintained individually in 1/2 L plastic containers until the fourth instar and afterwards in 1 L containers. Nymphs from instars 1–3 were fed *Drosophila nubin* ad libitum every other day, and afterwards with *Achaeta domesticus* crickets. The rest of the female mantises used were donated as adults by the Unidad de Manejo Ambiental Yolkatsin (México), where a colony of mantises raised in captivity has been maintained during several generations. These mantises were also fed *Drosophila* from instar 1–3, and *A. domesticus* afterwards. Thus, before our experiment, none of the mantises had been in contact with butterflies. All insects were maintained at ambient temperature under a 12 h dark–12 h light photoperiod in the insectary of the Instituto de Ecología (UNAM) located besides the PSAER.

The butterflies were randomly assigned to a treatment group: in the experimental group the hindwing tails were ablated (Fig. 1B), whereas in the control group the wings remained intact (Fig. 1A). Hindwing tails ablation was achieved by first introducing the butterflies in a −20 °C freezer until they were immobile (between 2 and 5 min), then the tails were cut out with micro-scissors (Iris Scissors; BioQp, Dominguez, CA, USA). Manipulation of each butterfly lasted approximately 2 min. Control individuals were also introduced in the freezer and manipulated for a similar amount of time as experimental butterflies. Twenty-six butterflies of both sexes were attacked thus producing experimental data (14 males: eight control, six experimental; 12 females: six control, six experimental; see Table A1).

Twenty-four female mantises were used, but five were never attacked. Twelve mantises that attacked were used just once (six with experimental and six with control butterflies) and seven were used twice (five first with a control and then with an experimental butterfly, and two first with an experimental and then with a control butterfly). Mantises used twice had a time interval between trials of at least two weeks thus reducing possible learning effects. The fact that only two of the seven mantises captured both butterflies and that other four captured the first but failed capturing the second butterfly, which suggests that learning had no effect on our results. To increase the probability of attack, mantises were starved three days before being exposed to a butterfly.

Butterflies were individually exposed to one mantis in a glass chamber measuring 29.5 cm × 25 cm × 9.5 cm (length × height × width), with one of the two largest (29.5 cm × 25 cm) sides covered with white Styrofoam. A Sony Handycam HDR-SR1was used to film most of the trials (23 out of 26). The room where the experiments were carried

out was illuminated with two 30 W white fluorescent tubes (Philips[TM] Slim line LDD F48T8/TL865) located at a diagonal distance from the chamber (i.e., they were not directly above it) of 2.5 m and 3.6 m, respectively. The mantis was introduced to the experimental chamber two hours before each trial. Afterwards, the butterfly was gently introduced in the chamber in a position as far as possible from the mantis. A trial was discarded if the mantis failed to attack the butterfly within 5 min. If the mantis attacked within five minutes after the introduction of the butterfly, we recorded the result (i.e., butterfly captured or escaped) and finished the trial. We allowed just one attack.

## RESULTS

We staged 22 control and 22 experimental interactions between a mantis and a butterfly. Twenty-six butterflies (59.1%) were attacked. The butterflies were attacked when they were walking, perching after walking or after landing; in one case the butterfly was detected after stepping on one leg of the mantis. The number of butterflies attacked (Fig. 2A) was statistically independent of the presence of hindwing tails (Chi squared = 0.38, $P = 0.54, gl. = 1$). The number of attacked butterflies displaying hindwing movements (that presumably allow the hindwing tails to mimic the movement of antennae) during the interaction with a mantis (Fig. 2B) was statistically independent of the presence of hindwing tails (Fisher's exact test, $P = 0.27$).

The number of butterflies surviving the attack (Fig. 2C) was statistically independent of the presence of hindwing tails (Fisher's exact test, $P = 0.70$). Attacks directed to the rear end of the butterfly resulted in less captures than those directed to other body parts (lateral and frontal attacks): five out of six butterflies escaped when attacked in the rear end, in contrast to four out of 17 attacks directed to other parts (Fisher's exact test, $P = 0.018$). (We have not videos of three interactions, one of them of a control butterfly that escaped.) However, two of the five failed attacks directed to the rear end involved butterflies with their hindwing tails ablated. Furthermore, only in one case the mantis directed the attack towards the "false head" despite the real head of the (control) butterfly was closer to the head and front legs of the mantis (see interaction between butterfly 127 and mantis 17 in seconds 27–43 of Video S1). In the other four failed attacks, the rear end of the butterfly was closer to the head and front legs of the mantis (see Video S1).

## DISCUSSION

In false head butterflies, the tails present in the anal angle of the hindwings are considered to mimic the antennae of the real head, a hypothesis consistent with the peculiar back-and-forth movements of the closed hindwings that apparently aid mimicking the movement of the real antennae (*Robbins, 1980*; *López-Palafox, Luis-Martínez & Cordero, 2015*). This idea led us to predict that the success in escaping a mantis attack would decrease in butterflies with "false antennae" experimentally ablated. Nevertheless, our experiment failed to reveal an advantage of possessing hindwings tails. The presence of hindwings tails in perching *C. xami* butterflies had no statistically significant effect on the probability of surviving an attack from a mantis that is possibly a natural predator.

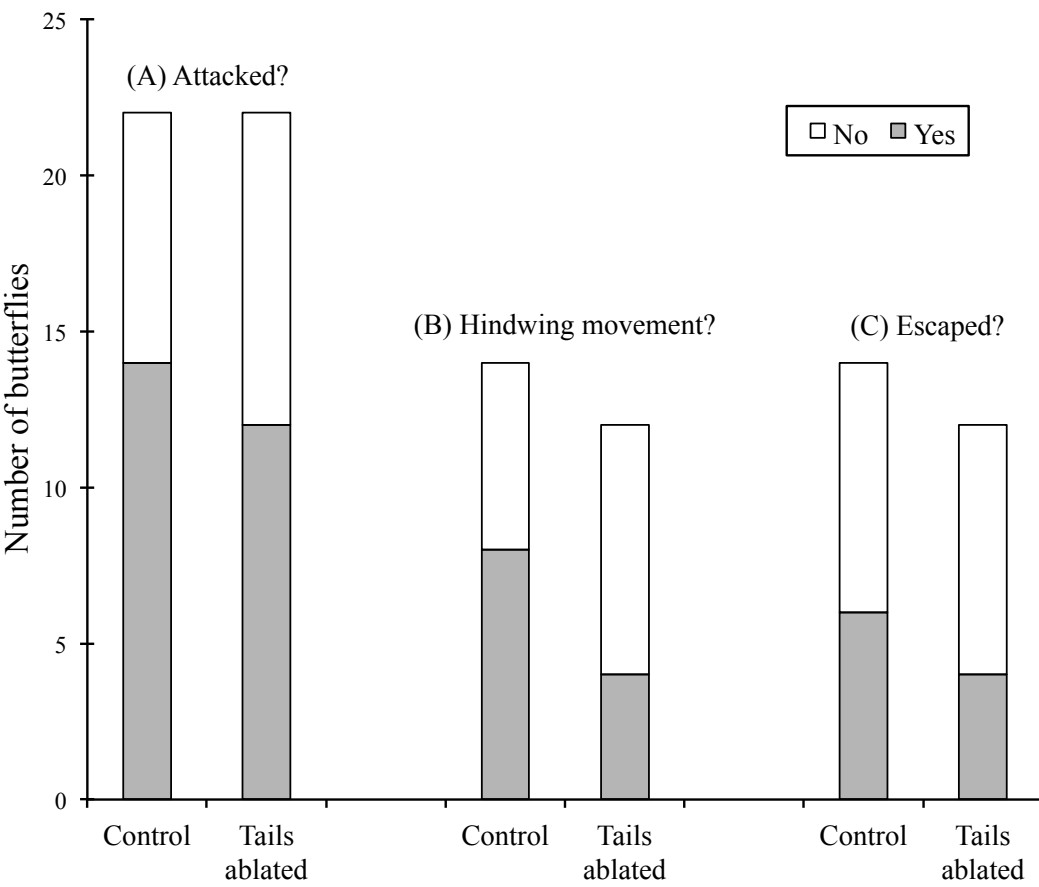

**Figure 2** **Experimental ablation of hindwing tails ("false antennae") in the "false head" butterfly *Callophrys xami* and its effect on interactions with female mantis (*Stagmomantis limbata*).** Control butterflies were manipulated in the same way as experimental butterflies but their hindwing tails were not ablated. (A) Number of butterflies attacked (gray) or ignored (white). (B) Number of butterflies that performed hindwing movements (gray) or not (white) before being attacked. (C) Number of butterflies escaping (gray) or being captured (white). None of the differences between control and experimental groups were statistically significant (see text).

Furthermore, although the absolute difference in the proportion of butterflies escaping an attack was in the predicted direction (Fig. 2), direct evidence of improved deception due to the presence of false antennae is weak. In one case, the mantis was apparently deceived into attacking the rear end (false head) of a butterfly with false antennae, judging from the fact that the mantis' head and front legs were closer to the real head (see interaction between butterfly 127 and mantis 17 in seconds 27–43 of Video S1). However, in the other four failed attacks directed to the rear end, the back of the butterfly was closer to the front legs of the mantis, suggesting that the mantis was not deceived into attacking that part. Furthermore, two of these four failed attacks involved butterflies with hindwing tails ablated.

There are several possible explanations for our results. First, hindwing tails could perform no function in this species, but being present because they were inherited from their phylogenetic ancestors. We cannot discard this possibility, but phylogenetic inertia seems unlikely considering that in Theclinae (the diverse subfamily including *C. xami*) false

head components evolve rapidly (*Robbins, 1981*). Second, hindwing tails could be involved in a different function, such as in courtship behaviour or flight manoeuvrability. These alternatives deserve further study. Finally, hindwing tails could improve the deceiving effect of ''false heads'' (i.e., act as ''false antennae'') against predators different from mantises, such as birds that detect their prey by using fine details of the wings and actively, and rapidly, approach it from a relatively long distance. In contrast, against a mantis, a predator that relies on crypsis and has a sit-and-wait strategy that allows more time to observe the prey at close range, hindwing tails could be useless. In fact, our observations suggest that *S. limbata* cryptic appearance and behaviour is quite successful against *C. xami* since in many cases the attacked butterflies approached the mantis (in one case was the butterfly was detected because stepped over a mantis leg). Furthermore, the back and forth movements of the closed hindwings, that presumably permit the ''false antennae'' mimic the movements of real antennae (*Robbins, 1980*; *López-Palafox, Luis-Martínez & Cordero, 2015*), possibly have a negative effect because they attract the attention of the mantis (*Prete et al., 2013*).

Although our observations show that in many cases mantises did not direct their attacks towards the ''false head'', and that many attacks resulted in successful capture of butterflies (16 out of 26 in our experiment), our study also indicates that at least some aspects of the ''false head'' help *C. xami* survive some mantis attacks, supporting the notion that they are adaptations against predators (*Robbins, 1980*; *Cordero, 2001*; *Sourakov, 2013*). Five out of six butterflies that were attacked in the ''false head'' zone were able to escape. In two of these cases (one control and one with hindwing tails ablated), the mantis teared small pieces of wing from the false head area (see interactions between butterfly 92 and mantis 16 in seconds 21–28, and between butterfly 129 and mantis X in seconds 59–62 of Video S1), an observation consistent with the idea that the ''false head'' area breaks-off easily (*Robbins, 1980*). Thus, our observations indicate that escaping from an attacking mantis depends on several factors, such as the ability to take flight rapidly (see Video S1) and the specific part of the wings grabbed by the mantis.

## APPENDIX

Table A1  **Raw data from the experiment on the effect of ablation of butterfly (*Callophrys xami*) hindwing tails (''false antennae'') on hindwing movement (HWM) and capture by female mantis (*Stagmomantis limbata*).** Control butterflies were manipulated in the same way as experimental butterflies but their hindwing tails were not ablated.

| Treatment | Mantis code | Butterfly[a] | HWM | Result |
|---|---|---|---|---|
| Control | 4 | 37-F | No | Captured[b] |
| Control | 11 | 14-M | No | Captured |
| Control | T | 106-F | No | Captured |
| Control | X | 117-F | No | Captured |
| Control | 1 | 1-M | Yes | Captured |

**Table A1** (*continued*)

| Treatment | Mantis code | Butterfly[a] | HWM | Result |
|---|---|---|---|---|
| Control | 11 | 126-M | Yes | Captured |
| Control | 15 | 90-F | Yes | Captured |
| Control | Z | 78-F | Yes | Captured[b] |
| Control | 5 | 39-M | No | Escaped |
| Control | 13 | 94-M | No | Escaped |
| Control | 1 | 38-M | Yes | Escaped[b] |
| Control | 12 | 86-M | Yes | Escaped |
| Control | 16 | 92-M | Yes | Escaped |
| Control | 17 | 127-F | Yes | Escaped |
| Tails ablated | 14 | 70-M | No | Captured |
| Tails ablated | 14 | 102-F | No | Captured |
| Tails ablated | 1E | 68-M | No | Captured |
| Tails ablated | 2E | 60-M | No | Captured |
| Tails ablated | A | 101-F | No | Captured |
| Tails ablated | T | 128-F | No | Captured |
| Tails ablated | 7 | 33-F | Yes | Captured |
| Tails ablated | 15 | 119-M | Yes | Captured |
| Tails ablated | 10 | 8-M | No | Escaped |
| Tails ablated | 6 | 42-F | No | Escaped |
| Tails ablated | 5 | 18-M | Yes | Escaped |
| Tails ablated | X | 129-F | Yes | Escaped |

**Notes.**
[a] M, male; F, female.
[b] Interaction not recorded in Video S1.

# ACKNOWLEDGEMENTS

This study is part of Tania Guadalupe López Palafox's (TGLP) Master in Sciences thesis in the Posgrado en Ciencias Biológicas, Universidad Nacional Autónoma de México. We thank Drs. Marcela Osorio, Atilano Contreras, Matthew Lin, Claudio Lazzari and Robert Robbins for valuable commentaries, and Raúl Martínez Becerril and Isabel Vargas Fernández for technical support. We thank Luis Antonio Cedillo Vázquez (UMA Yolkatzin) and Eric Martínez Luque for providing mantises. CC deeply loves María.

## Funding

This work was supported by PAPIIT/UNAM under grant IN210715 to Carlos Cordero. Tania Guadalupe López-Palafox was supported by a scholarship from CONACYT, México. The funders had no role in study design, data collection and analysis, decision to publish, or preparation of the manuscript.

## Grant Disclosures

The following grant information was disclosed by the authors:
PAPIIT/UNAM: IN210715.
CONACYT, México.

## Competing Interests

The authors declare there are no competing interests.

## Author Contributions

- Tania G. López-Palafox conceived and designed the experiments, performed the experiments, analyzed the data, wrote the paper, prepared figures and/or tables, reviewed drafts of the paper.
- Carlos R. Cordero conceived and designed the experiments, analyzed the data, contributed reagents/materials/analysis tools, wrote the paper, prepared figures and/or tables, reviewed drafts of the paper.

## Data Availability

The raw data is included in Table A1.

## Supplemental Information

Supplemental information for this article can be found online at http://dx.doi.org/10.7717/peerj.3493#supplemental-information.

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
