# Peer review of "Two-headed butterfly vs. mantis: do false antennae matter?"

_PeerJ, doi:10.7717/peerj.3493_

## Round 0.1 · original submission · Major Revisions

Both reviewers have appreciated your work and provided constructive criticism for improving the manuscript.

I would to add, that as for any biological interaction, the adaptive traits and the evolutionary history of a prey, cannot be understood without taking into account the characteristics of the predator. The attack of praying mantis is a classical issue in behavioural physiology since the 1950’s (e.g. Mittelstaedt Ann. Rev. Entomol. 1956) and an abundant literature is available, including recent contributions (e.g. Nityananda et al. Sci. Rep., 2015). In contrast to other predators hunting insects, as birds or jumping spiders, praying mantis perform extremely rapid ballistic attacks after computing the spatial coordinates of any object of the size of a potential prey; in addition, they are monochromatic. The question arise then about the choice of a predator that probably does not take into account the details of the prey aspect, for conducting prey-discrimination experiments. Unfortunately, not a single single article is cited on the subject, nor the capture strategy of the predator is discussed at all. It seems then difficult concluding about the evolution of prey traits, without considering the possibility that praying mantis could simply not be an appropriate models for testing the “false-head” hypothesis. This point is linked to a second one, i.e., the fact that conclusions are based on keeping the null hypothesis after testing a low number of individuals, considering that you are working with insects. On the one hand, keeping Ho means incertitude and not absence of difference. On the other hand, small sample sizes favorise keeping the null hypothesis.

In the opinion of this editor, these two elements, i.e., the predator strategy and the low sample size, must be taken into account in the interpretation of the results, in order to conclusions not be biased.

·

Basic reporting

no comment

Experimental design

no comment

Validity of the findings

no comment

Additional comments

The “false head” hypothesis posits that various aspects of wing pattern, wing shape, and behavior of adult lycaenid butterflies deflect the attack of visually hunting predators to the hind end of the butterfly, increasing the butterfly’s chances of surviving an attack.
Relevant observations in nature are rare. Van Someren (1922. J. E. Africa and Uganda Nat. Hist. Soc. 17: 18-21) anecdotally recorded the behavior of lizards attacking lycaenids, which was consistent with the false head hypothesis. Robbins (1981) reported that the incidence of an unsuccessful attack to the false head in the field was 5-6 times greater in species with “classic” false head wing patterns than in those with “regular” false head wing patterns, which make up the majority of Neotropical lycaenids. López-Palafox et al. (2015) showed that hindwing movements (a presumed false head behavior) increased when a stuffed insectivorous bird was placed near a butterfly in the field.
Experimental studies on the false head hypothesis are also rare. Perhaps the main reason is the difficulty in getting both the predator and the butterfly to behave normally. Building on the work of Sourakov (2013), López-Palafox and Cordero establish in this paper a new experimental protocol with female mantids as predators and Callophrys xami as the lycaenid prey in a small cage. Evidence that the butterflies are behaving “normally” is the occurrence of hindwing movements. Evidence for the mantids is that they attacked the butterfly in 26 of 42 instances. The authors carefully set up a protocol so that each of the 42 times that a mantis and butterfly were put together was an independent event. The primary variable was that the experimental butterflies had their tails (presumed false antennae) ablated. The amount of time and energy to gather 42 data points had to be prodigious. Kudos to the authors.
The main result is the difference in survival from an attack on butterflies with tails (6 out of 14, 42.9%) and with tails ablated (4 out of 12, 33.3%). The statistical results are clearly not significant (although a Fisher Exact Test might have been done because the expected value in one cell is less than 5), but might the statistically likely 9-10% increase in survival for butterflies with tails be biologically meaningful? Rhetorically, I might ask if the authors have sufficient demographic information on survival and adult lifespan of C. xami to explore this question.
The secondary result is that the authors made videos of the mantids attacking the butterflies. In a few cases, they note that attacks to the hind end of the butterfly allowed the butterfly to escape. These results are useful, albeit too few in number for robust conclusions. Building on the results in this paper, future work might use a lycaenid species with a more “classic” false head wing pattern, such as the Mexican Arawacus togarna. The contrasting results could be ground breaking, and it might be easier to obtain significant statistical results. For now, the current results are an important step in exploring the false head hypothesis.

·

Basic reporting

Generally fine with minor issues:
L50 "...a species of with false head..." (delete "of"?)

L53 "....because directed all its attacks..." (insert "it" after because?)

L75-79 is one single sentence without any commas. I suggest inserting a comma right after "...(Robbins, 1981), ...." at L76?

L91 I suggest spelling out the species name in full when used at start of a sentence?

L95 "Part of the females..." sounds wrong...I suggest "Some of the females..."

L119 dimensions should specify L x B x H (these info are missing

L123 Consider rephrasing this sentence? "A trial was discarded if a mantis failed to attack the butterfly within 5 min."

L125 consider rephrasing"(...i.e. butterfly captured or escaped)..."

L126-127 this entire sentence reads strangely...the authors might wish to edit and rephrase?
e.g. There was one exception, where the butterfly initially escaped but was eventually caught....

L183 "...an observation..." (delete the "s")

Experimental design

In M&M, the authors failed to mention clearly whether "rest of the mantis" donated by the Unidad de Manejo Ambiental Yolkatsin (L99-100) were given the same dietary care as the other females (that were the offspring of a female). The authors should highlight that none of the female mantises (in particular the ones that were donated) have had prior experience in handling butterflies.

L114. The reuse of predators in experiments usually complicates the statistics....but one way to control for this potential 'learnt' behaviour is to ensure a 'rest period', which I noted is ~2 weeks. It is probable that the mantises may have forgotten their 'learnt' behaviour of catching a butterfly, but perhaps the authors should seize this opportunity to mention the need for a two week time interval. It is obvious that this two weeks break relates to 'forgetting' the potential of learnt behaviour; the authors should perhaps highlight this point and cite any relevant references if possible?

L118 onwards ...
Since this is a visually-mediated experiment, there should be more information provided on the light environment. Is this experiment conducted in an open area using natural skylight (if so, do provide the timing and weather conditions if possible since cloudy and non-cloudy days, or direct sunlight, can mean that the colour contrast of the false antennae against the background can differ); if artificial lightings are used, are these full-spectral lights (about 300-700nm) or just normal fluorescent lights? Just some basic but accurate information on the nature of the light will suffice.

Validity of the findings

L190 If the mantises were given different diets or may have had prior experience in handling butterflies (which were not clearly stated in the M&M), it should perhaps be mentioned here?

Additional comments

This is a classic yet interesting project. Though the findings are not significant, the results do suggest that the false heads may potentially have 'distracted' the mantises. However, as this is mainly a visual experiment, the quality of the light environment may potentially play a big role; however there was no information provided on whether this predator-prey interaction was conducted under a full-spectrum (i.e. 300-700nm) light environment or not. There is a possibility that the false head (in particular the white colourations) may also reflect UV light, a visual signal that I am quite sure mantises are sensitive to. Now, the focus here should not be on UV reflection and UV vision, but I think the one main thing missing is the quality of the light environment. I strongly urge the authors to provide more detailed information on this and, depending on the description of the light environment in which the experiments were carried out, to provide more speculation on why the results were not significant.

---

## Round 0.2 · accepted · Accept

Thank you very much for the improvements introduced in the manuscript and for including the reviewers and myself in the acknowledgements. Congratulations for your nice work.

·

Basic reporting

no additional suggestions

Experimental design

no additional suggestions

Validity of the findings

no additional suggestions

Additional comments

The authors have done an excellent job of improving the manuscript. The video is especially excellent. I have no further modifications to suggest.